# A New Approach to Objectively Evaluate Inherited Metabolic Diseases for Inclusion on Newborn Screening Programmes

**DOI:** 10.3390/ijns8020025

**Published:** 2022-03-25

**Authors:** Alberto Burlina, Simon A. Jones, Anupam Chakrapani, Heather J. Church, Simon Heales, Teresa H. Y. Wu, Georgina Morton, Patricia Roberts, Erica F. Sluys, David Cheillan

**Affiliations:** 1Division of Inherited Metabolic Diseases, Reference Centre Expanded Newborn Screening, University Hospital Padova, 35128 Padova, Italy; alberto.burlina@unipd.it; 2Willink Biochemical Genetics Unit, Manchester Centre for Genomic Medicine, Manchester University NHS Foundation Trust, St Mary’s Hospital, Oxford Road, Manchester M13 9WL, UK; simon.jones@mft.nhs.uk (S.A.J.); heather.church@mft.nhs.uk (H.J.C.); hoiyee.wu@mft.nhs.uk (T.H.Y.W.); 3Department of Metabolic Medicine, Great Ormond Street Hospital NHS Foundation Trust, London WC1N 3JH, UK; anupam.chakrapani@gosh.nhs.uk; 4Neurometabolic Unit, University College London Hospitals NHS Foundation Trust and Enzymes Laboratory, Great Ormond Street Hospital NHS Foundation Trust, London WC1N 3JH, UK; simon.heales@gosh.nhs.uk; 5ArchAngel MLD Trust, Registered Charity No. 1157825, 59 Warwick Square, London SW1V 2AL, UK; georginamorton@archangel.org.uk (G.M.); patroberts.nbs@archangel.org.uk (P.R.); 6Helvet Health, Ruelle de la Muraz 4, 1260 Nyon, Switzerland; e.sluys@helvet-group.com; 7Service Biochimie et Biologie Moléculaire, Groupement Hospitalier Est, Hospices Civils de Lyon, 69002 Lyon, France

**Keywords:** newborn screening (NBS), inherited metabolic disease, inherited disorder, public health, paediatrics, rare diseases, genetics, congenital disorders, methodology, Wilson and Jungner

## Abstract

Newborn screening (NBS) programmes are essential in the diagnosis of inherited metabolic diseases (IMDs) and for access to disease modifying treatment. Most European countries follow the World Health Organisation (WHO) criteria to determine which disorders are appropriate for screening at birth; however, these criteria are interpreted and implemented by individual countries differently, creating disparities. Advances in research and diagnostics, together with the promise of new treatments, offer new possibilities to accelerate the expansion of evidence-based screening programmes. A novel and robust algorithm was built to objectively assess and prioritise IMDs for inclusion in NBS programmes. The Wilson and Jungner classic screening principles were used as a foundation to develop individual and measurable criteria. The proposed algorithm is a point-based system structured upon three pillars: condition, screening, and treatment. The algorithm was tested by applying the six IMDs currently approved in the United Kingdom NBS programme. The algorithm generates a weight-based score that could be used as the first step in the complex process of evaluating disorders for inclusion on NBS programmes. By prioritising disorders to be further evaluated, individual countries are able to assess the economic, societal and political aspects of a potential screening programme.

## 1. Introduction

Newborn screening (NBS) started in the 1960s as a critical part of public health programmes designed to test infants shortly after birth for disorders that can cause disability or death if left undetected and untreated [1]. The goal of NBS is thus to detect disorders before they become symptomatic [2]. While NBS includes a battery of tests, such as newborn hearing, this paper focuses solely on inherited diseases assessed by biochemical NBS carried out by dried blood spot (DBS) testing. 

In 1968, the World Health Organisation published a public health paper, “Principles and Practice of Screening for Disease”, that includes ten screening principles of early disease detection, commonly known as the Wilson and Jungner classic screening principles [3]. Since then, policymakers have been asked to consider an expanding list of disorders for inclusion in their national screening programmes due to advances in treatment options and in diagnostics, such as multiplex testing [4,5]. The Wilson and Jungner classic screening principles are widely used as guidelines for deciding whether a certain disorder is suitable for NBS [4]. However, as many aspects of the classic principles are subjective, disparities across and even within countries have arisen due to differences in the interpretation of the Wilson and Jungner screening principles. Considering Europe as a “geographical area consisting of around 50 countries situated east of the Atlantic Ocean, north of or in the Mediterranean Sea and west of the Ural Mountains, including all of Russia” [5], there are countries that screen newborns for over 20 disorders such as Italy, Hungary or Austria and others that screen newborns for as few as two; such as Armenia, Belarus, or Cyprus, [5]. There is a need for a systematic approach to evaluate disorders for inclusion on NBS programmes.

Our objective was to build an algorithm, based on the Wilson and Jungner classic screening principles, that generates a weight-based score for inherited disorders. The algorithm was designed to be the first step in the complex process of identifying disorders for inclusion on NBS programmes by allowing national authorities to objectively evaluate and prioritise inherited metabolic diseases (IMDs) for inclusion on NBS programmes. If a high score is calculated for a given disorder, it would then need to be evaluated at the local level to assess the economic, societal and political aspects of a potential screening programme. The proposed algorithm could limit the room for interpretation on which IMDs could be added to NBS programmes, reduce the disparity across European countries, and allow for horizon scanning of disorders for future consideration.

## 2. Materials and Methods

### Analysis of the 10 Classic Screening Principles

The Wilson and Jungner classic screening principles were used as the basis of our approach. We organised the ten screening principles into four categories; “Condition”, “Screening”, “Treatment” and “Other” (see Table 1).

Three of these categories contain principles that are both clinical and measurable: “Condition”, “Screening” and “Treatment”. The fourth category, “Other”, contains screening principles that are related to the economic, societal or political aspects of screening programmes. The principles that fell into the category “Other” were removed from the algorithm as they are neither clinical nor measurable. The emerging Wilson and Jungner criteria were also evaluated; however, these principles also fall into the category of “Other” as they encompass ethical, economic, and societal aspects of screening programmes which are not included in this algorithm [6]. The algorithm is therefore built upon three pillars: Condition, Screening and Treatment, encompassing the Wilson and Jungner screening principles 1, 2, 4–7 (see Table 2). If a disorder is highly-ranked by the algorithm, a country or regional assessment would be required to take into consideration the economic, societal and political aspects of screening programmes such as cost-effectiveness (principle 9), infrastructure (principle 3), patient-finding and treatment policies (principles 8 and 10). 

## 3. Results

### 3.1. Components of Novel Algorithm

With the intention of developing a point-based system, we explored the nuances of the six identified screening principles included in our algorithm, aiming to parse out individual criteria so that each factor could be assessed, and therefore, measured separately. It is worth mentioning that the Wilson and Jungner screening principles were not designed to evaluate disorders and therefore do not contain objective criteria. 

The three clinical and measurable pillars, Condition, Screening and Treatment, contain specific categories and criteria that directly measure the important aspects within each pillar. The weighting of the categories and criteria within the three pillars was determined through the testing of certain agreed upon disorders (see Appendix A, Table A1, Table A2, Table A3, Table A4, Table A5, Table A6, Table A7, Table A8, Table A9, Table A10, Table A11 and Table A12). The pillars and their respective categories and criteria are discussed in detail below.

#### 3.1.1. Pillar 1: Condition

This pillar considers Wilson and Jungner principles 1, 4, and 7; relating to the severity, onset and frequency of a disorder. Principle 1: “the condition should be an important health problem.” For this principle, we need to define and measure “important”, which relates to both the severity and the frequency of a disorder. Principle 4: “there should be a recognisable latent or early symptomatic stage”. This principle is related to the timing of the onset of the disorder. As the goal of NBS is pre-symptomatic detection, there must be a reasonable period of time in the natural history of the disorder during which symptoms are not present or not urgently demanding attention [3]. This implies that disorders with symptom presentation in the first few weeks of life are not as strongly recommended for NBS because patients would be diagnosed and treated as a result of early recognition and diagnosis [4]. However, one must keep in mind disorders that, despite presenting very early in life, are severely debilitating or fatal and have treatments which when administered pre-symptomatically provide superior outcomes. An example is maple syrup urine disease (MSUD), which presents with severe symptoms in the first two or three days of life and can cause irreversible brain damage and death if left untreated, but if identified prior to the development of symptoms, patients can achieve normal growth and development [7]. Principle 7 states that “the natural history of the disorder, including development from latent to declared disease, should be adequately understood”. This principle also underscores the importance of understanding and considering any genotype/phenotype correlations. As many IMDs have multiple phenotypes, or multiple forms, the algorithm must assess the severity, age of onset and proportion of patients with each form of the disorder. Therefore, for the pillar *Condition*, there are three categories: severity, onset, and frequency (see Table 3).

Severity: In looking at the severity of a disease, we must consider the progression and fatality for all forms of the disorder. While DBS tests may be able to identify affected individuals, some disorders have a broad spectrum of phenotypes, with both an early-onset, life-threatening form and an attenuated, less severe phenotype [8,9]. The proposed algorithm considers all forms (from mild to severe) of a given disorder and attributes 0.5 points to disorders with only severe forms, 0.5 points to disorders with a rapidly progressing form, and 1 point to disorders that can be fatal from birth to adolescence (see Table 3). These criteria are not exclusive, and a disorder can receive points for all components. For example, phenylketonuria (PKU) fulfils one criterion (see Table A11 in Appendix A), while medium-chain acyl-CoA dehydrogenase deficiency (MCADD) fulfils two (see Table A7 in Appendix A) and glutaric aciduria type 1 (GA1) fulfils all three criteria (see Table A1 in Appendix A).

Onset: Disease onset is one of the more complicated categories because many IMDs have multiple forms with different ages of onset for the different forms. There is agreement that disorders that are both early-onset and severe should be included in NBS, but there is not consensus that disorders that are adult-onset or have attenuated phenotypes should be included in NBS programmes [1]. Diagnosing late-onset disorders through NBS can create “patients in waiting” and could cause anxiety and uncertainty for healthcare practitioners and families in relation to prognosis, management and potential treatment options [1]. The proposed algorithm prioritises disorders where the larger majority of cases are early-onset in order to directly address access to early diagnosis and also take into account “patients in waiting” (see Table 3). To demonstrate how this would be applied we can compare two disorders: carnitine palmitoyltransferase (CPT) II deficiency and homocystinuria (HCU). CPT II has three subtypes: a lethal neonatal form, a severe infantile heptocardiomuscular form, and a myopathic form, which is the most common and where affected individuals have a normal life expectancy [10]. In HCU the majority of patients present with signs and symptoms during the first year of life, however there are also patients who do not develop symptoms until later childhood or adulthood [11]. While both disorders are asymptomatic for the first few weeks of life, the algorithm would prioritise HCU, where the majority of cases are early-onset, over CPT II, where the majority of cases are later-onset. Early-onset is defined per disorder based on the epidemiologic data available in the peer-reviewed literature. Although it is generally agreed that creating “patients in waiting” should be avoided, it is interesting to note that among individuals with later-onset lysosomal storage disorders (LSDs), the value of NBS for their disorder is recognised, as it would have reduced or eliminated their diagnostic odysseys and potentially altered their life planning [12].

Frequency: The algorithm assigns more points to disorders that have a higher frequency because NBS would allow for increased patient-finding. As disease frequency data is not always available, the algorithm considers either incidence or prevalence. There are four thresholds for frequency from ultra-rare (between 1 in 250,000 and 1 in 150,000) to more common (greater than or equal to 1 in 50,000; see Table 3). The highest-ranking threshold, for the more frequent disorders, includes disorders that are significantly less common as compared to those of the European Commission and the International Society for Pharmacoeconomics and Outcomes Research’s definition of “rare disease” [13].

#### 3.1.2. Pillar 2: Screening

This pillar considers Wilson and Jungner principles 5 and 6 relating to the availability and performance of diagnostic assays. Principle 5 states that “there should be a suitable test or examination”. According to Principle 6, “the test should be acceptable to the population”.

Availability: Since the 1970s, DBS have been routinely used in NBS because only a small volume of blood is required, and the sample is easily collected and transported on filter paper [13]. Tandem mass spectrometry (MS/MS) and liquid chromatography-tandem mass spectrometry (LC–MS/MS) have drastically advanced screening capabilities because one DBS can be analysed for an increasing number of metabolic disorders, allowing for the expansion of NBS programmes [5,14]. 

Performance: While the acceptability of a DBS test is generally accepted, the suitability of a test is subject to interpretation. We have defined this as the capacity of the DBS test to correctly identify patients with or without the disorder, or to put it another way, to have false positive and false negative rates that are acceptable for a screening test. Indeed, false-negative results allow true disease to go undetected, while false-positive results can induce a stressful situation and anxiety for families while waiting for confirmatory tests to be performed. Moreover, high false-positive rates result in a multiplication of confirmatory tests that can affect the cost-effectiveness of screening. There is not an agreed upon sensitivity, specificity, positive predictive value (PPV), or negative predictive value for screening tests in general [15]. In newborn screening, there can be additional complicating factors, such as the need to define referral rules based on levels of enzyme activity reported in the screening test [16].

The proposed algorithm recognises both the performance of the assay and its suitability as a screening test. Disorders which have a validated DBS assay with a lower false-positive rate are prioritised over disorders where a DBS test is in development, or a DBS test has a low performance. Additionally, if the DBS test has a high-false positive rate, low PPV, or if a tiered-screening strategy is required, the additional confirmatory strategy must be readily available to be integrated into the screening process (see Table 4). As an example, in mucopolysaccharidoses type I (MPS I), a two-tier screening process is in place, with confirmatory genetic testing carried out for newborns with abnormal alpha-l-iduronidase (IDUA) levels on a multiplex assay [17].

#### 3.1.3. Pillar 3: Treatment

This pillar considers Wilson and Jungner’s principle 2: “there should be an accepted treatment for patients with recognised disease.” Considering this principle and the need to add more granularity, we have created two categories for treatment: availability and outcomes.

Availability: Looking at treatment availability, the proposed algorithm ranks European Medicines Agency (EMA) approved treatments the highest, followed by other treatment interventions (such as diet, bone marrow transplant (BMT) or hematopoietic stem cell transplantation (HSCT)) or treatments in phase III clinical development (see Table 5). Treatments with EMA-approval have gone through rigorous clinical development programmes and received marketing authorisation (MA) for a specific indication [18]. Other treatment interventions, such as diet or HSCT, are less likely to have the same amount of published evidence or randomised controlled trials. The algorithm was designed to be agile and forward-looking; therefore, treatments in development are included to ensure that all available evidence is evaluated. Treatments in phase III clinical trials are assigned more points as compared with those in phase I/II clinical trials as they are more advanced in their likelihood to reach MA. The algorithm only considers one treatment strategy per disorder; the “most-advanced” treatment strategy, meaning that if there is an EMA-approved treatment available, there are no extra points given for other treatment interventions, or treatments in clinical development.

Outcomes: In addition to treatment availability, we must also evaluate patient outcomes. In the Treatment Outcomes category, the term “treatment strategy” includes EMA-approved treatments, treatments in development and treatment interventions, such as HSCT or BMT. In this category, treatment strategies that impact prognosis receive more points than those that only manage symptoms. Finally, disorders with a treatment strategy that can provide superior patient outcomes if initiated pre-symptomatically are strongly recommended independently of all other criteria (see Table 5). 

### 3.2. Weighting of Novel Algorithm

After all of the critical components were accounted for, we adjusted the relative weights, aiming to balance globally and within each pillar through the testing of certain disorders. A disorder can score a maximum of 13 points, six for *Condition*, three for *Screening* and four for *Treatment* (see Figure 1). *Condition* includes both the natural history and the frequency of a disorder, but with the natural history counting for twice as much, four points, as frequency which is only two points.

As in the Wilson and Jungner classic screening principles, the pillar *Condition* is the most comprehensive and therefore contributes to nearly half, 46%, of the total score in the algorithm (see Figure 2). The pillar *Treatment* contributes 31% of the total score, while the more straightforward pillar *Screening* contributes to 23% of the total score. The proposed algorithm prioritises severity of the disorder (23%) and treatment outcomes (20%), with other criteria contributing as follows to the total score: frequency of disorder (16%), availability of diagnostic test (15%), treatment availability (11%), performance of diagnostic test (8%) and onset of disorder (7%) (see Figure 2). 

### 3.3. Validation of Novel Algorithm

The algorithm was validated by applying the current IMDs that are on the UK NBS programme, assuming that these would be highly ranked by our algorithm. These disorders: GA1, HCU, isovaleric acidaemia (IVA), MCADD, MSUD and PKU are assessed individually with results shown in Figure 3 (detailed scoring in Appendix A, Table A1, Table A2, Table A3, Table 4, Table A5, Table A6, Table A7, Table A8, Table A9, Table A10, Table A11 and Table A12). Using the algorithm, these six disorders all scored above 8.5 points, therefore we propose that a disorder scoring ≥8.5 could be recommended for consideration for NBS programs after taking the economical and societal aspects into account.

## 4. Discussion

The proposed algorithm would provide an objective evaluation of the available evidence and generate a score to prioritise disorders for inclusion on NBS programmes. The algorithm does not include a cost effectiveness analysis, but instead facilitates the identification of disorders that would then need to be assessed in a health economics analysis phase. The algorithm was designed to include specific, individual criteria that could be used and applied to horizon scanning for future expansion of a NBS panel or when new information becomes available. For example, when a treatment receives EMA-approval, a disorder can be easily re-tested, and a new score can be determined. The algorithm is also flexible in its application and can be used to evaluate other inherited disorders. In this paper it has been applied to IMDs; however, it could be applied specifically to lysosomal storage disorders or to other inherited disorders as per specific country needs. Similarly, the algorithm could consider other major health and regulatory authorities, such as the Food and Drug Administration or the Medicines and Healthcare products Regulatory Agency in lieu of the EMA. 

While the algorithm was designed to be comprehensive, there are inherent limitations due to the complicated nature of inherited disorders. One limitation is that the algorithm only considers the highest scoring treatment per disorder. While this allows for an objective measure of the highest standard of care, disorders with multiple treatment options do not receive extra points. Secondly, as there are no guarantees for drug approval or successful outcomes of clinical trials, there is an inherent risk in horizon scanning for promising treatments. However, the algorithm assigns equal points to treatments in phase three development and treatment strategies because of the inherent complexities of completing robust phase three clinical trials in the rare disease space and the importance of this work. Lastly, while the algorithm aims to eliminate the subjectivity of NBS, there will always be ethical questions to consider. As an example, it is hard to imagine that there could be “an agreed policy on whom to treat as patients” as called for in the Wilson and Jungner principles. The algorithm was designed to prioritise disorders where most patients have an early-onset form; however, we must be aware that this also affects disorders where most patients have a later-onset form. For these disorders with a risk of late-onset, it is important to ensure that with NBS there are procedures in place to counsel families. The proposed algorithm attempts to balance the risk of creating patients-in-waiting for later-onset forms with the benefit of NBS for early-onset forms with rapid disease progression and a small treatment window.

Despite the rich discourse on how to come together across the EU and harmonise NBS programmes [5,19,20], there has been little forward motion. This algorithm attempts to provide a concrete tool for the progress that is so desired. The French National Authority for Health (Haute Autorité de Santé—HAS) has published their independent health technology assessment in 2020. The HAS developed an original method to prioritise diseases to be included in the existing neonatal blood screening program through a process of multi-criteria decision analysis (MCDA) using a number of criteria that are weighed by a limited number of stakeholders (physicians, labs performing the tests, patient groups and ethical experts). Specific diseases under assessment are subsequently scored on each of these criteria and ranked (article under review). This expertise allowed a recommendation to extend NBS by MS/MS to seven inborn errors of metabolism from the 24 studied (HCU, IVA, MSUD, TYR, GA1, Long-chain 3-hydroxyacyl-CoA dehydrogenase [LCHAD], and carnitine uptake defect/carnitine transport defect [CUD]). They will be implemented in the French NBS programme in addition to the two inborn errors of metabolism already screened [21]. This approach is a noteworthy model on how a horizon scanning tool, such as our proposed algorithm, could facilitate prioritization and expansion of NBS at a national level. In the United States we also see a model for evidence-based recommendations combined with regional implementation. The Recommended Uniform Screening Panel (RUSP) lists disorders which have passed scientific review and are recommended for universal screening; however, it is up to the individual states to decide which disorders will be screened for [22].

Expanded NBS programmes benefit patients, society, and the healthcare system. NBS programmes add to the current body of knowledge on the natural history of disorders because screening increases patient findings, and with more information on genotype/phenotype correlations, disease progression and treatment outcomes, further advances can be made. Our goal is for this algorithm to pave the way forward for evidence-based expansion of NBS programmes by allowing countries to objectively evaluate disorders while maintaining the ability to separately evaluate specific economic, societal, and political aspects of their own screening programmes. 

## Figures and Tables

**Figure 1 IJNS-08-00025-f001:**
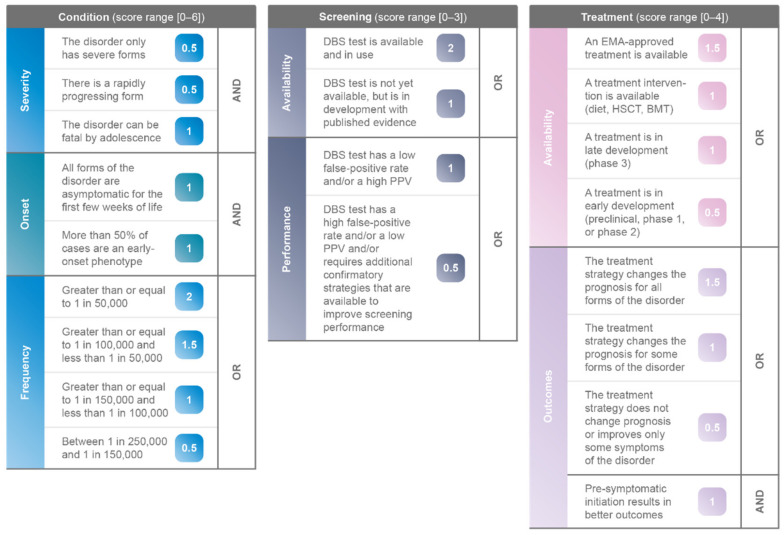
NBS evaluation algorithm.

**Figure 2 IJNS-08-00025-f002:**
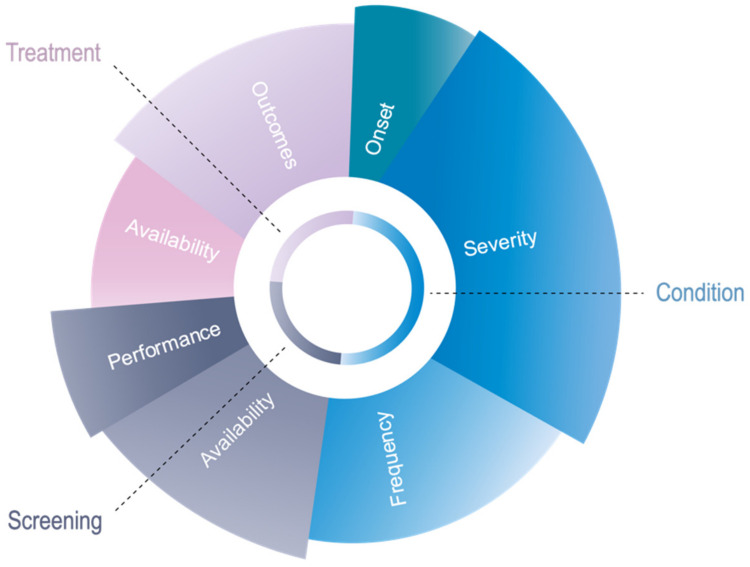
Components of novel algorithm.

**Figure 3 IJNS-08-00025-f003:**
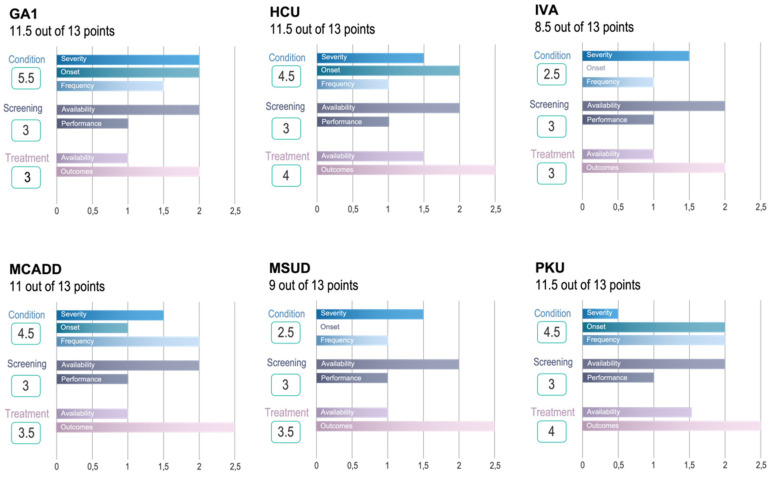
Total scores for the six IMDs on the UK newborn screening panel.

**Table 1 IJNS-08-00025-t001:** Wilson and Jungner classic screening principles organised into four key categories.

Wilson and Jungner Classic Screening Principles [3]		Category
1	The condition sought should be an important health problem	Condition
2	There should be an accepted treatment for patients with recognised disease	Treatment
3	Facilities for diagnosis and treatment should be available	Other
4	There should be a recognisable latent or early symptomatic stage	Condition
5	There should be a suitable test or examination	Screening
6	The test should be acceptable to the population	Screening
7	The natural history of the condition, including development from latent to declared disease, should be adequately understood	Condition
8	There should be an agreed policy on whom to treat as patients	Other
9	The cost of case-finding (including diagnosis and treatment of patients diagnosed) should be economically balanced in relation to possible expenditures on medical care as a whole	Other
10	Case-finding should be a continual process and not a “once and for all” project	Other

**Table 2 IJNS-08-00025-t002:** Three pillars of novel algorithm.

Pillar	Wilson and Jungner Classic Screening Principles
Pillar 1: Condition	Principle 1: The condition sought should be an important health problem
	Principle 4: There should be a recognisable latent or early symptomatic stage
	Principle 7: The natural history of the condition, including development from latent to declared disease, should be adequately understood
Pillar 2: Screening	Principle 5: There should be a suitable test or examination
	Principle 6: The test should be acceptable to the population
Pillar 3: Treatment	Principle 2: There should be an accepted treatment for patients with recognised disease

**Table 3 IJNS-08-00025-t003:** Components of Pillar 1, Condition.

Parameter.	Description	Score	Interaction
Severity	The disorder only has severe forms	0.5	AND
	There is a rapidly progressing form	0.5	
	The disorder can be fatal by adolescence	1	
Onset	All forms of the disorder are asymptomatic for the first few weeks of life	1	AND
	More than 50% of cases are an early-onset phenotype	1	
Frequency	Greater than or equal to 1 in 50,000	2	OR
	Greater than or equal to 1 in 100,000 and less than 1 in 50,000	1.5	
	Greater than or equal to 1 in 150,000 and less than 1 in 100,000	1	
	Between 1 in 250,000 and 1 in 150,000	0.5	

**Table 4 IJNS-08-00025-t004:** Components of Pillar 2, Screening.

Parameter	Description	Score	Interaction
Availability	DBS test is available and in use	2	OR
	DBS test is not yet available, but is in development with published evidence	1	
Performance	DBS test has a low false-positive rate and/or a high positive predictive value (PPV)	1	OR
	DBS test has a high false-positive rate and/or a low PPV and/or requires additional confirmatory strategies that are available to improve screening performance	0.5	

**Table 5 IJNS-08-00025-t005:** Components of Pillar 3, Treatment.

Parameter	Description	Score	Interaction
Availability	An EMA-approved treatment is available	1.5	AND
	A treatment intervention is available (diet, HSCT, BMT)	1	
	A treatment is in late development (phase 3)	1	
	A treatment is in early development (preclinical, phase 1, or phase 2)	0.5	
Outcomes	The treatment strategy changes the prognosis for all forms of the disorder	1.5	OR
	The treatment strategy changes the prognosis for some forms of the disorder	1	
	The treatment strategy does not change prognosis or improves only some symptoms of the disorder	0.5	
	Pre-symptomatic initiation results in better outcomes	1	AND

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
