# Peer review of "A New Approach to Objectively Evaluate Inherited Metabolic Diseases for Inclusion on Newborn Screening Programmes"

_2409-515X, 2022, doi:10.3390/ijns8020025_

Round 1
Reviewer 1 Report
Major Comments:
The paper describes a new approach to objectively evaluating inherited metabolic disorders for inclusion on newborn screening programs. The authors base their algorithms on categorizing Wilson and Jungner's principles and practice of mass screening for diseases. The introduction does not mention any other programs outside of Europe and the methodologies used by these programs to include disorders in their newborn screening programs (for example the RUSP as used in the USA). There should be more discussion about literature out there for other programs outside of Europe. They rightly point out the subjectivity inherent in the principles and the need for objective measures. They assign weights to different measures in the categories but there are no statistical analyses cited or performed that guided them as to how they came up with those weights used for the various subcategories. Scores are constrained to categories; for instance "Condition can accrue up to 6 points, "Diagnosis" up to 3 points and "Treatment" constrained to a maximum of 4 points. Need objective explanation given to the constraints imposed on scores for these categories. The authors tested their algorithm on 6 clearly defined metabolic disorders but did not test the algorithm on some of the less common disorders to see how they will perform. Finally the algorithm does not appear to be amenable to future disorders that could be considered for inclusion on newborn screening subcategories are assigned to weights without statistical support and total scores are constrained to find numbers.
Minor comment:
Line 227: the 2 sentences must be linked and not separate otherwise the first sentence will be incomplete.
Author Response
Review is in black bold, Responses are in blue
The paper describes a new approach to objectively evaluating inherited metabolic disorders for inclusion on newborn screening programs. The authors base their algorithms on categorizing Wilson and Jungner's principles and practice of mass screening for diseases. The introduction does not mention any other programs outside of Europe and the methodologies used by these programs to include disorders in their newborn screening programs (for example the RUSP as used in the USA). There should be more discussion about literature out there for other programs outside of Europe.
Our proposed algorithm indeed focuses on the EU, and it is not a literature analysis of other screening programs. If the algorithm is adopted by an international body or used at the country level it may need to be adapted. It is not set in stone, but rather a framework that is open to dialogue. We have, however, added in a comparison to the RUSP in the Discussion lines 332-336 as well as a comparison to a publication under review from the French National Health Authority, lines 319-329.
They rightly point out the subjectivity inherent in the principles and the need for objective measures. They assign weights to different measures in the categories but there are no statistical analyses cited or performed that guided them as to how they came up with those weights used for the various subcategories.
There is no precedent to draw on for utilizing statistical analysis in principles of newborn screening. This is an interesting suggestion for further investigation in another publication.
Scores are constrained to categories; for instance "Condition can accrue up to 6 points, "Diagnosis" up to 3 points and "Treatment" constrained to a maximum of 4 points. Need objective explanation given to the constraints imposed on scores for these categories. The authors tested their algorithm on 6 clearly defined metabolic disorders but did not test the algorithm on some of the less common disorders to see how they will perform.
The algorithm was validated on the conditions that are currently screened for in the UK. In the tandem publication, Jones et al. the algorithm was used on 48 IMDs some with a very low incidence, i.e. Farber disease where there are an estimated 200 cases worldwide.
Finally the algorithm does not appear to be amenable to future disorders that could be considered for inclusion on newborn screening subcategories are assigned to weights without statistical support and total scores are constrained to find numbers.
The algorithm is a horizon scanning tool, we are not sure to understand this actual comment.
Minor comment: Line 227: the 2 sentences must be linked and not separate otherwise the first sentence will be incomplete.
Thank you, this has been edited.
Additionally, Please note that we received the following comment for the tandem publication, IJNS-1502878, and therefore have updated this manuscript.
While perhaps not necessary to change for this manuscript, I would lean away from assigning one of the three pillars under the term "Diagnosis." Especially given that the criteria within that pillar address screening assay availability and performance within a DBS matrix and not the availability of diagnostic centers or testing.
Thank you, we agree, we will change the pillar “Diagnosis” to “Screening”. This has also been applied to both publications, IJNS-1502878 and IJNS-1502859.

Reviewer 2 Report
The paper by Burlina A et al. is trying to objectively evaluate inherited metabolic diseases for inclusion on newborn screening programmes. In the opinion of this reviewer before launching such an algorithm several questions that should be answered.
-They only evaluate “Condition”, “Diagnosis” and “Treatment”, and they do not evaluate the cost-effectiveness of a particular disease. I completely disagree, this is not objective, cost effectiveness must be included.
- Why a the test that is not yet available is graded?
- Why a treatment that is not yet available is graded? .
-Diseases with late-onset forms should have a negative punctuation, as it is.not desirable to diagnose such forms.
-Ethical problems should be discused in depth
Author Response
Review is in black bold, Responses are in blue
The paper by Burlina A et al. is trying to objectively evaluate inherited metabolic diseases for inclusion on newborn screening programmes. In the opinion of this reviewer before launching such an algorithm several questions that should be answered. -They only evaluate “Condition”, “Diagnosis” and “Treatment”, and they do not evaluate the cost-effectiveness of a particular disease. I completely disagree, this is not objective, cost effectiveness must be included.
There are many complex issues that have not been included in this algorithm. The algorithm is intended to be a horizon scanning tool to triage and propose conditions for further evaluation based on strong evidence from peer-reviewed literature. There is not enough evidence for early-stage analysis to accurately pull in cost-effectiveness in horizon-scanning. There is no standardization of cost-effectiveness as each system is different. After utilizing the algorithm, the next step would be to evaluate cost-effectiveness at the local level.
- Why a the test that is not yet available is graded?
The algorithm was designed for horizon scanning to evaluate all available evidence.
- Why a treatment that is not yet available is graded?
The algorithm was designed for horizon scanning to evaluate all available evidence.
-Diseases with late-onset forms should have a negative punctuation, as it is.not desirable to diagnose such forms.
This was considered in an earlier iteration, the authors decided on this scoring with the weighting of “more than 50% of cases are an early-onset phenotype” to account for this.
-Ethical problems should be discused in depth
The algorithm assigns points and was designed to objectively assess clinical and measurable components of newborn screening with peer-reviewed literature. Ethical considerations cannot be objectively assessed.
Additionally, Please note that we received the following comment for the tandem publication, IJNS-1502878, and therefore have updated this manuscript.
While perhaps not necessary to change for this manuscript, I would lean away from assigning one of the three pillars under the term "Diagnosis." Especially given that the criteria within that pillar address screening assay availability and performance within a DBS matrix and not the availability of diagnostic centers or testing.
Thank you, we agree, we will change the pillar “Diagnosis” to “Screening”. This has also been applied to both publications, IJNS-1502878 and IJNS-1502859.

Round 2
Reviewer 1 Report
Most concerns were addressed in the revised version. In figure 1, line 261 Diagnosis should change to Screening.
Author Response
All mentions of Diagnosis in the context of the proposed algorithm have been changed to Screening.
Please also note that:
- A precision was added on page 10 upon request from reviewer 2.
- An additional change was made in the abstract for consistency "three pillars: condition, screening and treatment."
- Other edits have been made to the authors affiliations, authors contribution and funding information but these do not impact the substance of the article.
Reviewer 2 Report
On page 11, please complete.,There are other seven conditions already screened in France. Therefore, you can say for example: In addition to the seven conditions already screened, seven other conditions by MS/MS have been included...................
Author Response
Review is in black bold, Responses are in blue
On page 11, please complete.,There are other seven conditions already screened in France. Therefore, you can say for example: In addition to the seven conditions already screened, seven other conditions by MS/MS have been included.
According to the French National Authority for Health (HAS) cited in reference number 21, there are 7 conditions screened for at birth of which only 2 are inborn errors of metabolism (PKU and MCAD) and can be screened with MS/MS. The other 5 conditions are: deafness, congenital adrenal hyperplasia, hypothyroidism, cystic fibrosis and sickle cell disease. These 5 conditions are not relevant to the current article and therefore have not been included.
Therefore, we have completed the paragraph on page 10 to meet your request: "This expertise allowed a recommendation to extend NBS by MS/MS to seven inborn errors of metabolism from 24 studied (HCU, IVA, MSUD, TYR, GA1, Long-chain 3-hydroxyacyl-CoA dehydrogenase [LCHAD], and carnitine uptake defect/carnitine transport defect [CUD]). They will be implemented in the French NBS programme in addition to the two inborn errors of metabolism already screened [21]."
Please note that:
- An additional change was made in the abstract for consistency "three pillars: condition, screening and treatment."
- Other edits have been made to the authors affiliations, authors contribution and funding information but these do not impact the substance of the article.